# Prediction of COVID-19 Using a WOA-BILSTM Model

**DOI:** 10.3390/bioengineering10080883

**Published:** 2023-07-25

**Authors:** Xinyue Yang, Shuangyin Li

**Affiliations:** School of Computer Science, South China Normal University, Guangzhou 510631, China; 20192434027@m.scnu.edu.cn

**Keywords:** COVID-19, infectious disease, BILSTM, WOA (whale optimization algorithm), prediction

## Abstract

The COVID-19 pandemic has had a significant impact on the world, highlighting the importance of the accurate prediction of infection numbers. Given that the transmission of SARS-CoV-2 is influenced by temporal and spatial factors, numerous researchers have employed neural networks to address this issue. Accordingly, we propose a whale optimization algorithm–bidirectional long short-term memory (WOA-BILSTM) model for predicting cumulative confirmed cases. In the model, we initially input regional epidemic data, including cumulative confirmed, cured, and death cases, as well as existing cases and daily confirmed, cured, and death cases. Subsequently, we utilized the BILSTM as the base model and incorporated WOA to optimize the specific parameters. Our experiments employed epidemic data from Beijing, Guangdong, and Chongqing in China. We then compared our model with LSTM, BILSTM, GRU, CNN, CNN-LSTM, RNN-GRU, DES, ARIMA, linear, Lasso, and SVM models. The outcomes demonstrated that our model outperformed these alternatives and retained the highest accuracy in complex scenarios. In addition, we also used Bayesian and grid search algorithms to optimize the BILSTM model. The results showed that the WOA model converged fast and found the optimal solution more easily. Thus, our model can assist governments in developing more effective control measures.

## 1. Introduction

SARS-CoV-2 is a highly infectious virus that spreads through direct, aerosol, and contact transmission [1]. Infected individuals experience symptoms such as a fever, dry cough, and runny nose, and in severe cases, breathing difficulties, pneumonia, and death. As of 20 December 2022, COVID-19 had claimed 31,431 lives and infected 9,482,570 individuals in China [2]. The virus has caused great harm to human life and health, and its impact extends beyond public health to affect the world’s economies and healthcare systems [3]. In January 2020, The World Health Organization even declared COVID-19 a global pandemic.

In order to make adequate preparations and make better prevention measures before each epidemic outbreak, how to accurately predict the number of new infections in a certain area has become a necessary problem for people to solve.

An accurate prediction of the number of new infections in a certain area has become a necessary problem to solve in order to make adequate preparations and make better prevention measures, such as beefing up containment measures in advance, which can prevent outbreaks from spreading. Additionally, accurate predictions can help the medical system to prepare its physical and human resources, ensuring that they are equipped to handle any potential influx of patients. By making use of accurate predictions, governments can optimize their response plans and allocate resources more efficiently, ultimately saving lives and resources. It is essential to prioritize accurate predictions of infectious diseases, as they can be the key to mitigating and even preventing their spread.

To make accurate predictions about COVID-19, many scholars have proposed novel coronavirus prediction models, which can be classified into three categories: mathematical models, machine learning models, and deep learning models. Mathematical models include Arima and propagation dynamics models (SI, SIR, SIRS, SEIR, etc.). Machine learning models include SVM, Lasso, and linear regression. Common deep learning models include CNN, LSTM, BILSTM, GRU, and TCN.

In the propagation dynamics models, the population can be divided into four categories: S, E, I, and R. S means a susceptible person. E means an exposed person. I means an infectious person. And R means a recovered person. By combining these four groups of people according to different infectious diseases, we can produce the SI, SIR [4,5,6,7], SIRS [8], and SEIR [9,10,11,12] models. Among these models, SIR and SEIR are more suitable for COVID-19 prediction and are used by most people for COVID-19 prediction. In 2020, Moein et al. [6] conducted simulations of the epidemic using SIR in Isfahan province, Iran, between 14 February and 11 April and predicted the trajectory of the outbreak for the remaining period by considering three scenarios with varying degrees of social distancing measures. They found that while SIR could make short-term predictions, it was difficult to predict the long-term epidemic spread, which was also reflected in other studies that used SIR to make predictions. Compared with the SIR model, the SEIR model adds the concept of an exposed person, which is more in line with the characteristics of COVID-19 and has a more reliable result. In 2020, Zhong Nanshan’s team [12] came up with a forecasting model that combined urban migration data with SEIR. By using urban migration data to adjust the numbers for S, E, I and R, they predicted the development of the epidemic in Zhejiang, Guangdong, and Hubei, which explained the impact of different control methods on the epidemic. This model can well simulate the development trend of an epidemic. For example, it can predict the peak period and decline period of the epidemic in a certain period so that the control policy can be formulated in advance. However, the model is greatly affected by the infection rate of the population and the degree of isolation of the region. In addition, the model curve is very smooth. Therefore, it cannot accurately predict the infection situation of specific people in the region on a certain day.

The autoregressive integrated moving average (ARIMA) model is the most commonly used time series prediction model, which can predict outbreaks in the next few days. It first uses logarithmic and difference methods to stabilize the data. Then, it determines the model parameters *q* (the autoregressive parameter) and *p* (the moving average order) using autocorrelation functions (ACFs) and partial autocorrelation functions (PACFs). Using the values of *p* and *q*, we can invoke the ARIMA model to make predictions. In 2020, Yang et al. [13] combined data from the period when the outbreak of new infections in Wuhan reached zero to establish an ARIMA model and used it to make predictions for Italy, which had a similar situation. After the experiments, they found that the ARIMA model was better for making short-term predictions. Roy et al. [14] used overlay analysis to classify India into very high-, high-, medium-, and low-risk zones for COVID-19 and applied the ARIMA model to make a prediction for India. They concluded that only data from a time series can deduce linear relationships. This approach did not work well for events that can be influenced by multiple factors, including several meteorological and specific social influences. Furthermore, the method cannot be applied to other diseases. Haneen et al. [15] also used the ARIMA model for prediction. Their experimental data came from Kuwait and the $R^{2}$ value of the experiment was as high as 0.996, demonstrating the high accuracy of the model. However, they also stated that this model was not suitable for predicting sudden outbreaks or for use in complex environments.

Traditional machine learning models [16,17,18,19] have also made great contributions to predicting infectious diseases. Rustam et al. [20] used LR (linear regression), Lasso (least absolute shrinkage and selection operator), SVM (support vector machine), and ES (exponential smoothing) to calculate the number of newly infected cases, deaths, and recoveries in the next 10 days of some states/provinces of Australia, Canada, Algeria, and Afghanistan. The experimental results showed that in all data sets, ES made the best prediction, followed by LR, Lasso, and SVM. Rath et al. [21] proposed the use of a multiple linear regression model to predict the daily number of active cases by including daily positive cases, recoveries, and deceased cases as input variables. They compared the multiple linear regression model with the traditional linear regression model and concluded that the former performed better.

Recently, deep learning models have been applied to the prediction of time series problems, such as temperature prediction [22] and stock prediction [23], which have achieved good results. Given the strong correlation between COVID-19 and time, many researchers have used deep learning models such as RNN, LSTM, BILSTM, CNN, GRU, and some hybrid models [24,25,26,27,28,29] to predict COVID-19 cases. For example, Xu et al. [29] used CNN, LSTM, and CNN-LSTM models to predict COVID-19 cases in Brazil, India, and Russia and found that the LSTM model performed the best among the three models. Jin et al. [30] integrated TCN, GRU, DBN, Q-learning, and SVM to create a TCN-GRU-DBN-Q-SVM model for predicting the next day’s cases in India, the United Kingdom, and the United States. Mohimont et al. [31] designed a CNN model that can predict short-term confirmed and hospital cases using a small amount of data. Moreover, they designed a TCN (temporal convolutional network) model, which can predict confirmed cases, hospitalizations, artificial ventilation hospitalizations, and recoveries with good accuracy. Shahid et al. [32] proposed forecast models comprising ARIMA, SVR, LSTM, Bi-LSTM, and GRU. They came to the conclusion that model ranking from good performance to the lowest in entire scenarios were Bi-LSTM, LSTM, GRU, SVR, and ARIMA. Gautam et al. [33] proposed a model that combined transfer learning with the LSTM model. By training the model using the data from Italy, the United States, and other countries, they applied the trained model to make single-step and multi-step predictions for Germany, France, Brazil, India, and Nepal. The results showed that even in the face of different intervention policies, the model still achieved good prediction accuracy. Considering that the LSTM-RNN model was not accurate enough for prediction, Natarajan et al. [24] proposed an RNN-GRU model to predict infections, recoveries, and deaths in four countries (the Czech Republic, the United States, India, and Russia).

Some researchers proposed combining Internet data with epidemic data to make predictions. Ayyoubzadeh et al. [34] used linear regression and LSTM models to make predictions for Iran based on Google Trends data. Although the prediction effect was not ideal, they proposed a prediction method based on network information. Guo et al. [35] proposed the WCT (Weibo COVID-19 trends) model, which was built using a dataset of Weibo posts from users in Wuhan and based on the logistic regression model. This model improved upon the shortcomings of the GFT (Google Flu Trends) model, which tended to overestimate the peak of the epidemic.

Furthermore, some scholars have employed intelligent optimization algorithms in combination with neural networks for optimizing neural network model parameters. An et al. [36] proposed a BILSTM model based on the attention mechanism, which was optimized using the sparrow optimization algorithm. They applied this model to make predictions for Egypt, Ireland, Iran, Japan, and Russia and achieved good results. Prasanth et al. [37] utilized specific search term data from Google Trends related to the COVID-19 pandemic, along with COVID-19 spread data from the European Centre for Disease Prevention and Control (ECDC), to make predictions. In their prediction model, they used the grey wolf optimizer (GWO) to optimize the LSTM model. Compared with the ARIMA model, their model exhibited better accuracy.

However, in the predictive models for COVID-19, most scholars do not use optimization models to optimize the parameters of these models. Even if scholars do use the optimization model, important factors such as the time step and optimizer selection are ignored for optimization. Moreover, the input data features used by these models are often too simple, relying solely on past cumulative new cases to predict future cases. As a result, these models are highly accurate only for short-term predictions. When we use them to predict data several days later, the accuracy drops dramatically. Furthermore, the training effectiveness of these models will also be greatly reduced when future outbreaks become complex. Therefore, there are many places where these neural network models can be optimized, which could significantly improve the accuracy of model prediction.

In Shahid’s article [32], his team came to the conclusion that the models ranked from good performance to the lowest in entire scenarios were Bi-LSTM, LSTM, GRU, SVR, and ARIMA. Given Shahid’s research results and the shortcomings of these deep learning models, we used BILSTM as our basic model and enhanced it with the following modifications. Our approach involved enriching the characteristics of input data, including cumulative new cases, cumulative recoveries, cumulative deaths, existing infections, daily new cases, daily recoveries, and daily deaths. Additionally, we utilized the optimization algorithm WOA (whale optimization algorithm) to optimize the model parameters and increase the number of optimization parameters compared with other optimization models. These measures could significantly enhance the accuracy of the predictions.

In our experiments, we used MAE, RMSE, MAPE, and R2 as evaluation indicators. Furthermore, using the epidemic data from Guangdong, Chongqing, and Beijing, we compared our model with several recognized baselines, including LSTM, BILSTM, GRU, CNN, CNN-LSTM, RNN-GRU, DES (double exponential smoothing), ARIMA, linear, Lasso, and SVM. The experimental results showed that our model outperformed the other baselines in terms of prediction accuracy. Our model was also able to handle complicated situations, highlighting its robustness and versatility. Moreover, we explored alternative approaches, such as Bayesian and grid search algorithms, to optimize and compare their performance instead of the WOA model. The experiments indicated that the running time of the WOA model was not excessively long, and its optimized accuracy was the highest. This finding further validated the claim that the WOA model converges quickly and efficiently finds the optimal solution.

We hope that our model can provide useful insights and assist the government in formulating effective measures for epidemic prevention and control.

## 2. Methods and Models

### 2.1. LSTM and BILSTM

LSTM is a variant of the RNN model that is commonly employed for processing time series data. It was developed to address the issue of an RNN’s difficulty in long-term learning and dependency. The central concept of LSTM lies in its memory cell with a gated function. Its gated system consists of three gates: input gate ($i_{t}$), output gate ($O_{t}$), and forget gate ($f_{t}$). Figure 1 illustrates the structure of LSTM.

The input gate ($i_{t}$) controls the input of the current information. When the input information passes through the unit, the input gate will perform a calculation to determine whether to input the current information. The memory gate ($f_{t}$) controls whether to retain past information. When the past information passes through the unit, the memory gate performs a calculation to determine whether to retain the information. The output gate ($O_{t}$) controls the output of the current information. It determines whether to output the current information by performing a calculation. Additionally, $C_{t}$ represents a long-term memory unit, while $h_{t}$ represents a short-term memory unit. The specific calculation processes are as follows:(1)$$f_{t}=\sigma (w_{f}\cdot \left[h_{t-1},x_{t}\right]+b_{f}$$
(2)$$i_{t}=\sigma \left(w_{i}\cdot \left[h_{t-1},x_{t}\right]+b_{i}\right)$$
(3)$$C_{t}^{\prime }=tanh\left(w_{c}\cdot \left[h_{t-1},x_{t}\right]+b_{c}\right)$$
(4)$$O_{t}=\sigma \left(w_{o}\cdot \left[h_{t-1},x_{t}\right]+b_{o}\right)$$
(5)$$C_{t}=f_{t}\cdot C_{t-1}+i_{t}\cdot C_{t}^{\prime }$$
(6)$$h_{t}=O_{t}\cdot tanh\left(C_{t}\right)$$

In the formulas above, σ is the sigmoid activation function. The variables w and b in the formula denote the weight and intercept, respectively.

BILSTM is a variant of LSTM (Figure 2) that incorporates an additional layer of reverse calculation alongside the base LSTM. As shown in Figure 2, the original sequence is ($A_{0}$, $A_{1},\;A_{2},\ldots ,A_{i}$), while the reversed sequence is represented as ($A_{0}^{\prime }$, $A_{1}^{\prime },\;\;A_{2}^{\prime },\ldots ,A_{i}^{\prime }$). The final output value is determined by the forward sequence and the reverse sequence:(7)$$y_{i}=v_{1}\cdot A_{i}+v_{2}\cdot A_{i}^{\prime }$$

In this formula, $v_{1}$ and $v_{2}$ represent the corresponding weights associated with the two sequences. In some scenarios, BILSTM trains better than LSTM, and this held true for the problem at hand.

### 2.2. WOA

The whale optimization algorithm is an intelligent optimization algorithm proposed by Mirjalili [38] in 2016. It was inspired by the preying behavior of whales and aims to adjust parameters to discover the optimal solution. As a metaheuristic optimization algorithm, it relies on straightforward concepts and is easy to implement. Furthermore, it exhibits fast convergence and can bypass local optima easily.

Similar to other metaheuristic optimization algorithms, this algorithm first generates an initial population and calculates the fitness value of each individual. Then, it traverses the current population to find the individual with the best fitness. After that, it updates the location of individuals in the population by imitating the behavior of whales, including encircling prey, bubble-net attacking, and searching for prey. As the population progresses to the next generation, it continues searching for the optimal individual and updating individual positions until the maximum number of iterations is reached.

The values of *P* and *A* determine the manner in which an individual whale updates its position. The mathematical model is as follows:(8)$$P=random\left(0,1\right)$$
(9)$$\;R_{1}=random\left(0,1\right)$$
(10)$$a=2-2*t/T_{max}$$
(11)$$A=2\;*\;a\;*\;R_{1}\;–\;a$$

Among them, *t* is the current number of iterations and $T_{max}$ is the maximum number of iterations.

In addition, there are several factors below that affect the update of location:(12)$$\;R_{2}=random\left(0,1\right)$$
(13)$$C=2*\;R_{2}\;$$
(14)$$l=random\left(-1,1\right)$$

When *P* < 0.5 and |*A*| < 1, the whale individual updates its position by encircling prey. The calculation formula is below, where $X\left(t\right)$ is the position of the current individual, $X\left(t+1\right)$ denotes the updated position of the individual, and $X_{best}$ represents the current optimal individual:(15)$$D_{1}=|C*X_{best}-X\left(t\right)|$$
(16)$$X\left(t+1\right)=X_{best}-A*D_{1}$$

When *P* < 0.5 and |*A*| ≥ 1, the whale individual will search for prey. During this process, the whale randomly selects positions to force itself away from the prey, thereby enabling global search. This can be expressed as follows:(17)$$rand=randint\left[1,whale\_num\right]$$
(18)$$D_{2}=|C*X_{rand}-X\left(t\right)|$$
(19)$$X\left(t+1\right)=X_{rand}-A*D_{2}$$

And when *P* ≥ 0.5, it updates its position by bubble-net attacking. Meanwhile, the whale approaches its prey in a spiral motion to capture its food. Therefore, it can be expressed using the following formula:(20)$$D_{3}=|X_{best}-X\left(t\right)|$$
(21)$$X\left(t+1\right)=X_{best}+D_{3}*\;e^{b*l}*cos\left(2\pi l\right)$$
where $D_{3}$ represents the distance between the current individual and the optimal individual. And *b* is a coefficient that represents the shape of the whale’s spiral, which was set to 1 here.

The pseudocode for the WOA algorithm is shown in Algorithm 1.
**Algorithm 1.** The pseudocode of the WOA algorithm.**Input:**G: the maximum iterationsb: a constant for defining the shape of the logarithmic spiral*n*: the number of whale populations**Output** : Optimal individual Xbest and its fitness value fg1: Initialize the whale population Xii=1,2,…,n2: Calculate the fitness value of each individual3: **while** (t < G)4: **for** each individual5: Update a, A, C, l, P (some constants)6: **if1** (*P* < 0.5)7: **if2** (|*A*| < 1)8: Xi updates its position by encircling prey9: **else if2** (|*A*| ≥ 1)10: Xi updates its position by searching for prey11: **end if2**12: **else if1** (*P* ≥ 0.5)13: Xi updates its position by bubble-net attacking14: **end if1**15: **end for**16: Check to see if any individuals are out of range and remove them if they are17: Calculate the fitness value of each individual18: Update the current optimal individual Xbest  and its fitness fg19: **end while**20: Return Xbest, fg

### 2.3. WOA-BILSTM

Given the strong correlation between the novel coronavirus outbreak and time series, we opted to use the BILSTM neural network model for its ability to handle timing issues effectively. Additionally, we employed the whale optimization algorithm (WOA) to optimize the model parameters. With these enhancements, our proposed WOA-BILSTM model achieved high accuracy in predicting the cumulative number of confirmed cases several days in advance (Figure 3).

In the first step of our model, data preprocessing plays a crucial role in improving the prediction accuracy. To enhance the performance of the prediction models, we incorporated additional input features compared with other models. These input features included cumulative confirmed cases, cumulative cured cases, cumulative death cases, existing cases, daily confirmed cases, daily cured cases, and daily death cases in a province or municipality.

For a specific location, the variable “daily confirmed cases” represents the number of newly confirmed cases reported within a given day. The variable “daily cured cases” denotes the number of individuals who have recovered from the disease on a daily basis. Similarly, the variable “daily death cases” indicates the number of deaths recorded daily due to the disease. “Existing cases”, also referred to as “active cases” or “current cases”, represents the number of individuals actively infected with the disease at a particular point in time. “Cumulative confirmed cases”, “cumulative cured cases”, and “cumulative death cases” refer to the cumulative sums of “daily confirmed cases”, “daily cured cases”, and “daily death cases”, respectively, since the beginning of the epidemic. These variables provide a comprehensive overview of the COVID-19 situation. Hence, we employed these variables to make predictions about COVID-19.

In addition, to prepare the data for training and testing, it was normalized and divided into training and testing sets in a proportion of our choice. The data from consecutive days were used as the input to train the model and predict infection rates for the upcoming days.

In the second step, a neural network model was designed for the algorithm. Our neural network model consists of two layers of BILSTM and one fully connected layer. To prevent overfitting, dropout is applied after each BILSTM layer. The activation function used is linear.

The last step is to adjust the BILSTM parameters for optimal results. We used WOA to optimize some parameters in the BILSTM model, enabling us to obtain the best combination of parameters. To ensure a stable result, the BILSTM model with the same parameter set is evaluated ten times, and the R2 value of the resulting predictions is averaged. After obtaining the best parameters, we input them into the BILSTM model to make predictions.

Through these steps, we can make accurate predictions about the number of COVID-19 infections in the future.

### 2.4. Evaluation Parameters

In this study, we employed MAE, RMSE, MAPE, and $R^{2}$ as evaluation metrics, which are commonly used as an assessment of COVID-19 predictions. These metrics are described as follows, where *n* represents the number of samples, $y_{i}$ represents the true value of sample *i*, and $y_{i}^{\prime }$ represents the predicted value of sample *i*.

MAE is the average absolute error, representing the average absolute error between the predicted value and the real value. A smaller MAE indicates a better model. The formula for MAE is as follows:(22)$$\mathrm{MAE}=\frac{1}{n}\sum_{i=1}^{n}\left|y_{i}-y_{i}^{\prime }\right|,\in \left[0,+\infty \right)$$

MSE is the mean square error, representing the error between the predicted value and the real value. RMSE is the root mean square error, which is the square root form of MSE. The closer the value of RMSE is to 0, the better.
(23)$$\mathrm{MSE}=\frac{1}{n}\sum_{i=1}^{n}{\left(y_{i}-y_{i}^{\prime }\right)}^{2},\in \left[0,+\infty \right)$$
(24)$$\mathrm{RMSE}=\sqrt{\mathrm{MSE}},\in \left[0,+\infty \right)$$

MAPE is the percentage absolute error. When MAPE is close to 0, it is a perfect model. When MAPE is greater than 1, it is a bad model.
(25)$$\mathrm{RMAPE}=\frac{100\%}{n}\sum_{i=1}^{n}\left|\frac{y_{i}-y_{i}^{\prime }}{y_{i}}\right|,\in \left[0,+\infty \right)$$

$R^{2}$ is the correlation coefficient, indicating the degree of agreement between the predicted data and the real data. The closer the value is to 1, the better the model effect will be.
(26)$$R^{2}=1-\frac{\sum_{i=1}^{n}{\left(y_{i}-y_{i}^{\prime }\right)}^{2}}{\sum_{i=1}^{n}{\left(y_{i}-\bar{y}\right)}^{2}},\in \left(-\infty ,1\right]$$
where $\bar{y}$ is the real value of the average.

## 3. Experiments

We wrote the code for the network model in Python, which was constructed in the TensorFlow 2.12.0 framework. The experimental hardware used was a GPU NVIDIA GeForce GTX 1650, with an energy consumption of 183.96 W·h.

The data came from the website https://news.sina.cn/zt_d/yiqing0121 (20 February 2023), which is the official data of China’s Health Commission. Through the website, we obtained data for Guangdong, Chongqing, and Beijing. Due to the significant variations in COVID data year by year, we only used one year’s worth of data (Table 1), which went from 20 December 2021 to 20 December 2022.

In the experiments, sets 0–229 of data (20 December 2021–6 August 2022) were used for training and sets 230–330 of data (7 August–14 November 2022) were used for testing. For the input values, each set of data contained several days’ worth of information and each day’s information contained seven cases (cumulative confirmed cases, cumulative cured cases, cumulative death cases, existing cases, daily confirmed cases, daily cured cases, daily death cases). And the output value of the model is the cumulative number of new cases after a few days. Thus, it is a multiple parallel input and single-step output model.

For example, if the time-step is 7, then the first set of data was from 20 December 2021 to 26 December 2021 and the predicted date was 27 December 2021 (Figure 4).

To evaluate the performance of our model, we compared it with other neural network models, namely, LSTM, BILSTM, GRU, CNN, CNN-LSTM, and RNN-GRU. In these models, we set the dropout to 0.01, the time-step to 7, and the optimizer to Adam. And the input data for them was the cumulative number of new cases in the previous period. For other parameters, we used a grid search for the optimization. As shown in Figure 5, we predicted the cumulative number of confirmed cases after 1 day, 5 days, and 7 days based on previous data. However, since the time step is a variable, the forecast time range for these three areas may vary slightly.

According to the real data, the overall epidemic changes were similar in these three places. This was because Guangdong, Chongqing, and Beijing are all parts of China that are governed by similar epidemic prevention policies.

Before November, the growth rates of the epidemic in these three places remained relatively stable, resulting in minimal changes in the growth rates of both the training set and the test set. Hence, all models produced accurate forecasts before November. But after November, the epidemic rates in these three cities began to increase rapidly, coinciding with a major outbreak in China. As a result, almost all of the models exhibited worse predictions during this period.

However, it can be seen from the overall prediction curve that the WOA-BILSTM model always had the best prediction effect. Furthermore, with the increase in prediction time, our model still has good accuracy.

Moreover, we used R2, MAE, RMSE, and MAPE to evaluate the models (Table 2). The results showed that the R2 values of most models were more than 0.9 when projecting one day into the future. Surprisingly, the R2 values of our model were above 0.993 in this case. All models except our model had very low R2 values when projecting after five and seven days, while the R2 values of our model were still higher than 0.9. Therefore, we could conclude that our model predicted much better than other neural network models.

Furthermore, we conducted an additional experiment to compare our model with DES, ARIMA, linear, Lasso, and SVM models. In this experiment, we employed our model, as well as the other models, to predict the future for seven consecutive days, specifically from 27 August to 2 September 2022. It is worth mentioning that the training values of the other models were from 20 December 2021 to the day before the forecast. Again, we used MAE, RMSE, and MAPE to evaluate the models’ performances.

To study the correlation of the time series itself, we first performed ACF and PACF analyses of the training sets of each place, as shown in Figure 6. Based on the ACF and PACF charts, we can see that the data was not seasonal. This also showed that it is sufficient to use only ARIMA and DES (double exponential smoothing) methods for prediction, rather than SARIMA and triple exponential smoothing methods. Moreover, the parameters for ARIMA were also determined according to the ACF and PACF charts.

Figure 7 displays the true values and the predicted values of these models. From Figure 7, we can infer that our model, the DES model, and the ARIMA model exhibited the closest match between the predicted and true values. However, the linear, Lasso, and SVM models had a large gap between the predicted and real values. Especially in the prediction of Guangdong province, the results of the linear, Lasso, and SVM models were more than 200 different from the real value. Moreover, in the prediction of Beijing, the SVM model produced particularly poor results.

Table 3 presents the predicted values. In Table 3, we can see that our model, the DES model, and the ARIMA model all produced excellent results for Guangdong and Beijing, with MAPE values of less than 0.01. When predicting Chongqing, the MAPE values of the WOA-BILSTM, DES, and ARIMA models were 0.0128, 0.0239, and 0.0131, respectively. In addition, in the prediction of Guangdong and Chongqing, the accuracy of the WOA-BILSTM model was the best, followed by the ARIMA and DES models. However, in the prediction of Beijing, the ARIMA model was slightly better than our model.

Why was our model a bit worse than the ARIMA model when we did the experiment from 27 August to 2 September? To find out the reason, we did the experiments for two other periods in Beijing (Table 4). Similar to previous experiments, our model achieved the best prediction effect in the two periods.

Figure 8 shows the real data for Beijing from 1 August to 12 September. According to the real data, there is little change in the epidemic trend in the period from 27 August to 2 September and the period before 27 August. However, the data from 13 August to 19 August and from 6 September to 12 September showed significant changes. This also corroborated other people’s findings that ARIMA is not suitable for complex situations. In contrast, our model produced excellent results in both general and complex cases.

In addition, to show the energy consumption of our model, we also used Bayesian and grid search methods to optimize the parameters of the improved BILSTM model. Based on the previous data, the experiment conducted a one-day forecast for the Beijing area from 14 August to 20 November 2022.

For the WOA optimizer, we set the population size to 10 and the number of iterations to 5. The time step value ranged from 4 to 14. The first and second layers of BILSTM ranged from 1 to 100. The dropout value ranged from 0.01 to 0.5. The batch_size ranged from 1 to 128. The learning rate value was selected from 0.1, 0.01, 0.001, 0.0001, 0.00001, and 0.000001. The optimizer was chosen from SGDS, Adagrad, Adadelta, RMSprop, Adam, Adamax, and Nadam. In order to make the adjusted model more stable, each prediction was the average of ten runs, which was also applied to grid optimization and Bayesian optimization. The optimal parameter set after WOA optimization was {time step: 4, the first layer of BILSTM: 87, the second layer of BILSTM: 34, dropout: 0.366, batch_size: 1, optimizer: Adam, learning rate: 0.0001}.

For the Bayesian optimizer, we set the init_points to 10 and n_iter to 5. The parameters range was consistent with the WOA algorithm. Its optimal parameter set was {time step: 11, the first layer of BILSTM: 32, the second layer of BILSTM: 68, dropout: 0.419, batch_size: 3, optimizer: Nadam, learning rate: 0.0001}.

For the grid search algorithm, in order to reduce the number of iterations, the time step value ranged from 7 to 14. The first and second layers of BILSTM were selected from (16,32,64). The batch_size value was selected from (32,64,128). The learning rate was set to 0.001. The dropout value was set to 0.3 and the optimizer was Adam. After the iterations, the optimal parameter set was {time step: 7, the first layer of BILSTM: 16, the second layer of BILSTM: 64, dropout: 0.3, batch_size: 32, optimizer: Adam, learning rate: 0.0001}.

The specific operation time and results of these models are shown in Table 5. According to the tabular data, the grid search model ran the longest but was less accurate than the WOA model. The Bayesian model ran for 44 min and its results were not bad. And the WOA model had the highest accuracy, with a running time of 2 h and 24 min, which is acceptable. This shows that the WOA had a fast convergence rate and the best optimization effect.

## 4. Conclusions

In this paper, we propose a WOA-BILSTM model. For the model, we took cumulative confirmed cases, cumulative cured cases, cumulative death cases, existing cases, daily confirmed cases, daily cured, and daily death cases as inputs. Then, we used the WOA to train some parameters of the BILSTM model. Furthermore, we compared it with some other models, such as LSTM, BILSTM, GRU, CNN, CNN-LSTM, RNN-GRU, DES, ARIMA, linear, Lasso, and SVM. The experimental results showed that our model had the highest accuracy in predicting regions in China. When compared with the deep learning models, our model had the best prediction results and its prediction effect was significantly improved compared with other models. And when compared with the Arima and machine learning models, our model was robust and accurate. Therefore, our model has a good prediction effect and universal applicability in COVID-19 prediction research. Additionally, we also used Bayesian and grid models instead of the WOA model for optimization. By comparing the accuracy of the model and the uptime, we found that the WOA model was the most accurate while ensuring that the uptime was not too long. This indicated that the WOA model converged faster and could find the optimal solution more easily. Thus, our proposed WOA-BILSTM model is more suitable for COVID-19 prediction. We hope that our model can help the government make better prevention and control measures during the epidemic.

## Figures and Tables

**Figure 1 bioengineering-10-00883-f001:**
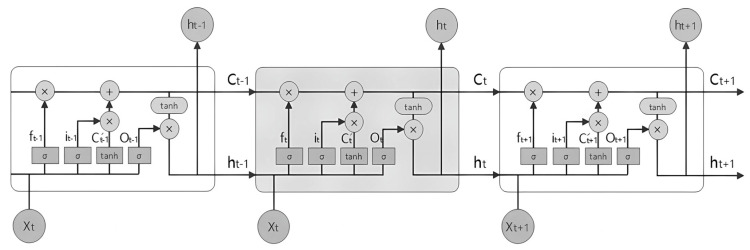
The structure of LSTM.

**Figure 2 bioengineering-10-00883-f002:**
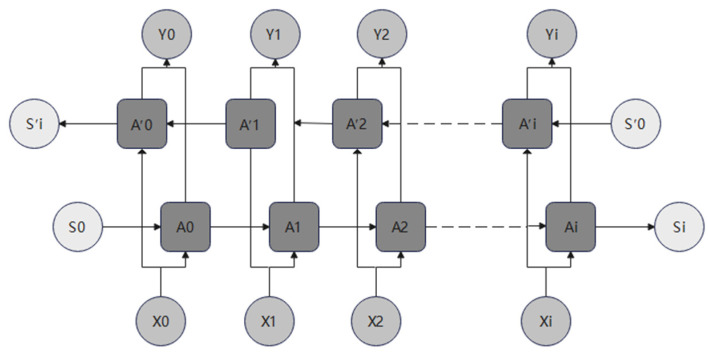
The structure of BILSTM.

**Figure 3 bioengineering-10-00883-f003:**
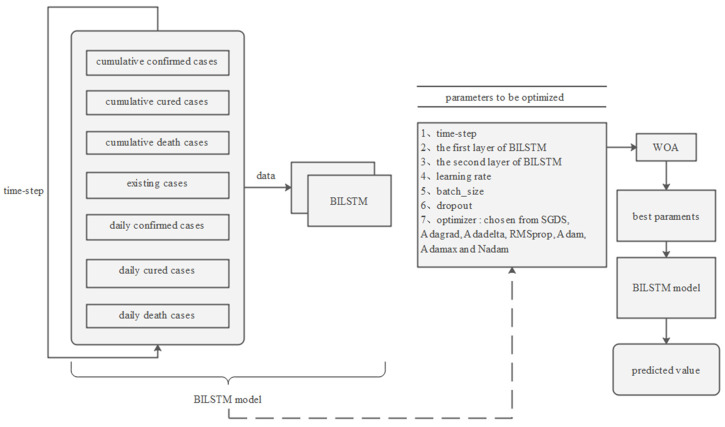
The structure of the WOA-BILSTM model.

**Figure 4 bioengineering-10-00883-f004:**
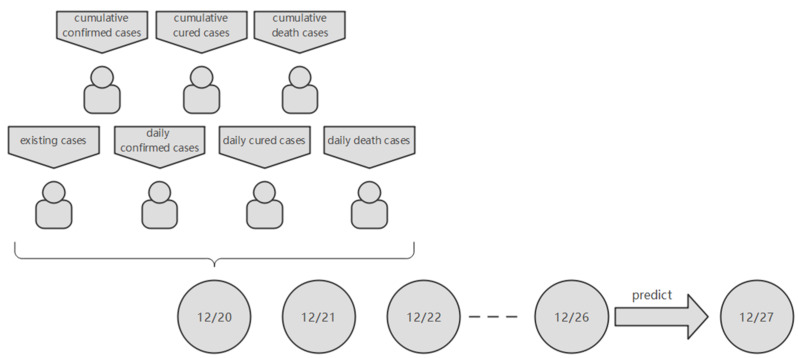
When the time-step is 7 and the forecast day is 1 day later, the input of the model is shown in the figure.

**Figure 5 bioengineering-10-00883-f005:**
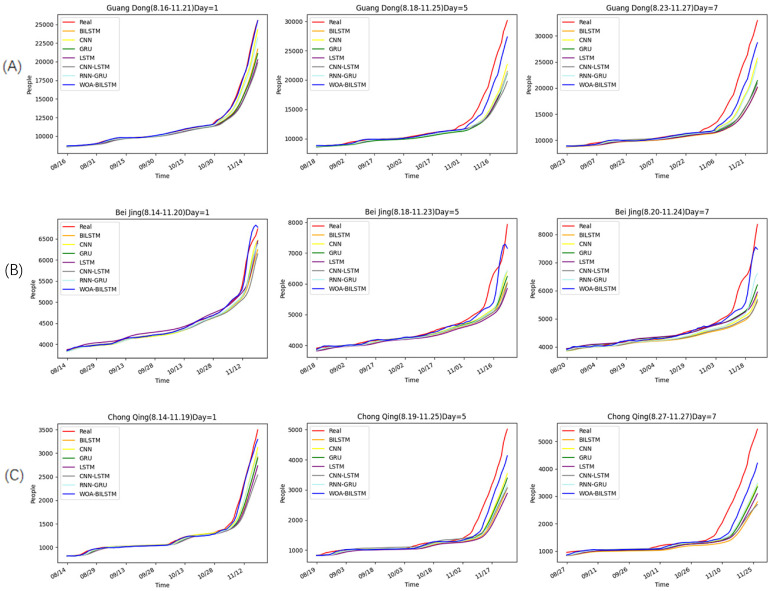
Forecasts of the outbreak in three places in the next 1, 5, and 7 days. (**A**) The prediction for Guangdong. (**B**) The prediction for Beijing. (**C**) The prediction for Chongqing.

**Figure 6 bioengineering-10-00883-f006:**
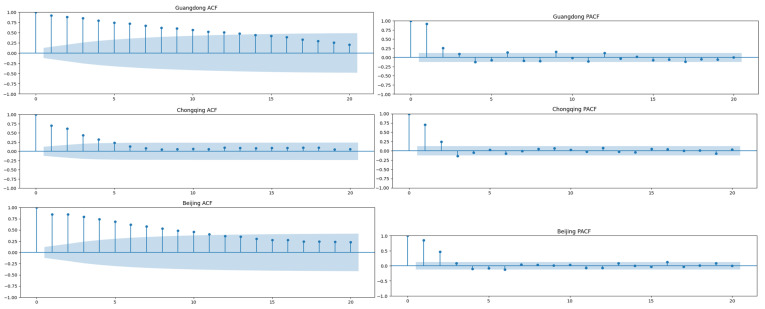
ACF and PACF diagrams of Guangdong, Chongqing, and Beijing.

**Figure 7 bioengineering-10-00883-f007:**
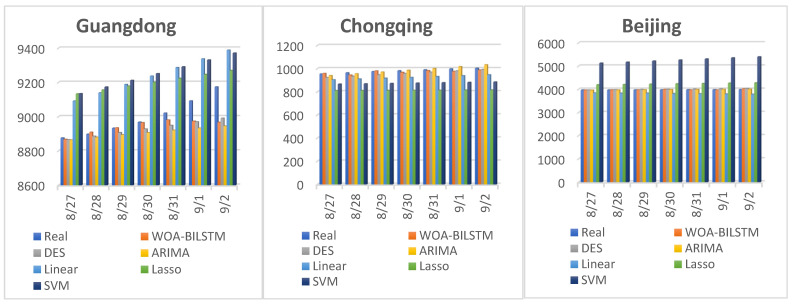
We used WOA-BILSTM, DES, ARIMA, linear, Lasso, and SVM models to forecast Guangdong, Chongqing, and Beijing. The figure shows the predicted values and true values.

**Figure 8 bioengineering-10-00883-f008:**
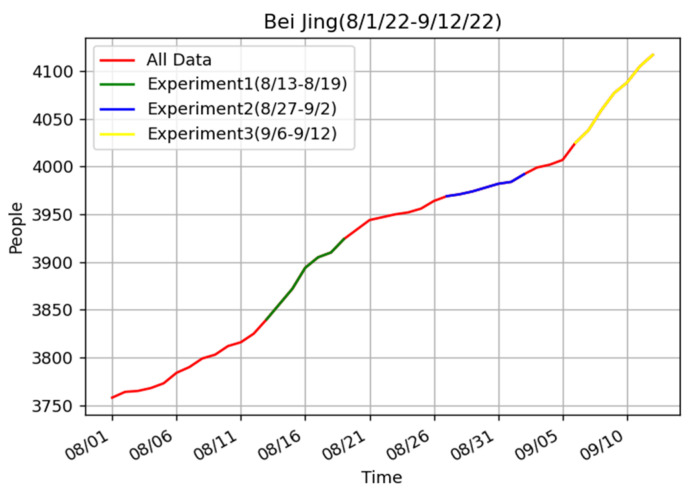
The real data of Beijing from 1 August to 12 September 2022.

**Table 1 bioengineering-10-00883-t001:** Data from 20 December 2021 to 20 December 2022.

	Province/Municipality	20 December 2021	21 December 2021	…	20 December 2022
**Cumulative** **Confirmed Cases**	Guangdong	3394	3399	…	62,367
	Chongqing	610	610	…	9972
	Beijing	1205	1205	…	28,389
**Cumulative** **Cured Cases**	Guangdong	3305	3310	…	52,359
	Chongqing	602	602	…	7435
	Beijing	1180	1180	…	16,762
**Cumulative** **Death Cases**	Guangdong	8	8	…	8
	Chongqing	6	6	…	7
	Beijing	9	9	…	20
**Existing Cases**	Guangdong	81	81	…	10,000
	Chongqing	2	2	…	2530
	Beijing	16	16	…	11,607
**Daily Confirmed Cases**	Guangdong	10	5	…	1189
	Chongqing	0	0	…	205
	Beijing	1	0	…	544
**Daily Cured Cases**	Guangdong	6	5	…	993
	Chongqing	0	0	…	76
	Beijing	1	0	…	360
**Daily Death Cases**	Guangdong	0	0	…	0
	Chongqing	0	0	…	0
	Beijing	0	0	…	0

**Table 2 bioengineering-10-00883-t002:** Data from 20 December 2021 to 20 December 2022.

		Day 1				Day 5				Day 7			
Area	Model	MAE	RMSE	MAPE	R^2^	MAE	RMSE	MAPE	R^2^	MAE	RMSE	MAPE	R^2^
Guangdong	WOA-BILSTM	78.54	122.71	0.006	0.9988	579.38	1141.06	0.032	0.9449	877.00	1701.81	0.046	0.9084
	LSTM	642.32	1414.70	0.038	0.8467	1216.67	2651.49	0.063	0.7027	1881.98	3848.48	0.093	0.5314
	BILSTM	503.61	1111.82	0.030	0.9053	1213.30	2618.00	0.063	0.7102	1954.32	3862.76	0.100	0.5279
	GRU	584.83	1228.40	0.036	0.8844	1383.83	2742.47	0.077	0.6819	1726.29	3542.38	0.085	0.6030
	CNN	272.30	530.28	0.018	0.9785	1131.30	2377.87	0.060	0.7609	1243.50	2562.68	0.061	0.7922
	CNN-LSTM	722.80	1512.55	0.044	0.8248	1332.84	2934.48	0.068	0.6359	1657.67	3537.28	0.079	0.5960
	RNN-GRU	334.80	730.20	0.020	0.9592	1205.65	2599.51	0.063	0.7142	1371.89	2766.80	0.068	0.7578
Chongqing	WOA-BILSTM	20.95	38.55	0.013	0.9951	131.14	260.13	0.056	0.9190	104.14	214.23	0.018	0.9278
	LSTM	96.11	213.51	0.048	0.8512	290.76	588.70	0.120	0.5851	234.32	528.87	0.039	0.6531
	BILSTM	80.77	177.07	0.041	0.8976	261.92	544.58	0.105	0.6449	326.10	641.32	0.056	0.4899
	GRU	80.03	173.96	0.041	0.9012	217.06	456.03	0.086	0.7510	217.44	487.01	0.036	0.7058
	CNN	57.25	121.99	0.030	0.9514	201.85	422.44	0.080	0.7863	276.85	580.61	0.046	0.5819
	CNN-LSTM	106.94	246.55	0.052	0.8015	240.66	513.03	0.096	0.6849	365.14	772.11	0.126	0.4841
	RNN-GRU	64.46	139.15	0.033	0.9368	233.89	497.23	0.091	0.7040	302.91	617.32	0.108	0.6702
Beijing	WOA-BILSTM	19.97	52.97	0.004	0.9933	82.88	198.36	0.015	0.9409	104.14	241.23	0.018	0.9278
	LSTM	82.03	157.61	0.016	0.9403	261.22	494.72	0.047	0.6325	234.32	528.87	0.039	0.6531
	BILSTM	83.29	185.33	0.015	0.9174	220.30	442.87	0.038	0.7055	326.10	641.32	0.056	0.4899
	GRU	69.94	149.80	0.013	0.9460	190.97	400.41	0.033	0.7592	217.44	487.01	0.036	0.7058
	CNN	74.98	139.57	0.014	0.9532	171.74	368.27	0.029	0.7963	276.85	580.61	0.046	0.5819
	CNN-LSTM	102.15	226.73	0.019	0.8764	213.63	458.16	0.036	0.6848	306.16	625.79	0.051	0.5142
	RNN-GRU	71.15	154.63	0.013	0.9425	188.12	366.47	0.033	0.7983	229.37	453.46	0.039	0.7450

**Table 3 bioengineering-10-00883-t003:** Prediction from 27 August 2022 to 2 September 2022.

Area	Guangdong			Chongqing			Beijing		
Evaluation	MAE	RMSE	MAPE	MAE	RMSE	MAPE	MAE	RMSE	MAPE
WOA-BILSTM	55.792	91.645	0.0061	12.615	14.047	0.0128	10.068	13.553	0.0025
DES	66.591	89.555	0.0074	22.697	23.669	0.0239	20.127	23.705	0.0050
Arima	88.178	115.416	0.0097	12.939	15.404	0.0131	8.250	9.651	0.0021
Linear	243.793	244.654	0.0271	56.145	56.226	0.0572	160.857	163.058	0.0404
Lasso	206.825	214.382	0.0231	168.079	168.735	0.1711	256.217	257.044	0.0644
SVM	257.509	259.049	0.0287	105.516	106.063	0.1074	1284.986	1287.661	0.3229

**Table 4 bioengineering-10-00883-t004:** Prediction of Beijing from 6 September 2022 to 12 September 2022 and from 13 August 2022 to 19 August 2022.

Time	6 September 2022–12 September 2022			13 August 2022–19 August 2022		
/	MAE	RMSE	MAPE	MAE	RMSE	MAPE
WOA-BILSTM	40.690	47.003	0.0100	30.912	34.401	0.0079
DES	45.023	50.214	0.0112	71.667	74.373	0.0188
Arima	52.293	57.656	0.0128	38.021	40.487	0.0098
Linear	264.215	269.782	0.0648	46.237	55.688	0.0118
Lasso	260.405	260.446	0.0640	196.927	196.979	0.0507
SVM	1210.207	1211.450	0.2971	5143.148	5143.441	1.3235

**Table 5 bioengineering-10-00883-t005:** R2 values and runtime of the WOA, Bayesian, and grid search models.

Model	Tenth Mean (R2)	Maximum Value (R2)	Time
WOA-BILSTM	0.9905	0.9948	2 h 24 min
Bayes-BILSTM	0.9716	0.9888	46 min
Grid-BILSTM	0.9838	0.9913	5 h 8 min

## Data Availability

The data came from the website https://news.sina.cn/zt_d/yiqing0121 (accessed on 20 February 2023).

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
