# Peer review of "Prediction of COVID-19 Using a WOA-BILSTM Model"

_bioengineering, 2023, doi:10.3390/bioengineering10080883_

Round 1

Reviewer 1 Report

Title: "Prediction of Covid19 using WOA-BILSTM model"
Authors: Xinyue Yang  and Shuangyin Li
School of Computer Science, South China Normal University,
Guangzhou 510000, China.

----------------------------

Summary:

The article proposes a Whale Optimization Algorithm-Bidirectional Long Short-Term
Memory (WOA-BILSTM) model for predicting cumulative confirmed cases of COVID-19
The researchers use regional epidemic data as input to the model, which includes
cumulative confirmed, cured, and death cases, and existing cases and daily confirmed,
cured, and death cases. After initial training of the model the WOA algorithm is
used to optimize specific parameters. The model is tested on the epidemic data from
Beijing, Guangdong, and Chongqing in China from 2021 to 2022 and compared with the
LSTM, BILSTM, GRU, CNN, ARIMA, Linear, Lasso, and SVM models. The results indicate
the the proposed model outperforms these alternatives and achieves higher accuracies
in complex scenarios.

----------------------------

General Comments:

I think that this article is of interest to the readers of MDPI
Bioengineering. It addresses an important topic of predicting the spread of
COVID-19 and similar epidemics from the past data which may be helpful to
public and private agencies in devising defensive measures to combat the
spread of epidemics.

The article should be published after a number of modifications
that I address in my specific comments below. The authors must
answer the following three questions in their revised version:

1) How much data were given to their model for training (in gigabytes)?

2) What hardware facilities (GPU architectures, the number of
GPUs, and the types of GPUs) were used for training and testing?

3) How long did the training and testing took in terms of physical time (hours)
and energy (kW-h)?

Researchers and practitioners who have been using DL methods are coming to
the conclusion that these models are not just data-hungry. They are also,
and that may be even more fundamental in the long run, energy-hungry. They
achieve better accuracies because they overparameterize the problem feature
space. But that methodology has a cost: enormous time and energy footprints.
Hence, my three questions above. In many real-world situations, less
accurate models may be preferable to more accurate models if the latter have
significantly larger energy footprints.

----------------------------

Specific Comments:

1) Lines 34-36: In order to make adequate preparations and make better
prevention measures before each epidemic outbreak, how to accurately
predict the number of new infections in
a certain area has become a necessary problem for people to solve.

This is not a grammatically correct sentence. Please rephrase it to read
as follows:

Accurate prediction of the number of new infections in a ... has  become
a necessary problem to solve in order to make preparations ...

2) The Related Work Section should be omitted. The relevant work should
be mentioned in the Introduction and then critiqued and discussed in the
Discussion Section. There is no need to have a separate section for it.
This is a recent and growing trend in many peer-reviewed journals published
in English.

3) Lines 172-174: Whale optimization algorithm is an intelligent optimization
algorithm proposed by Mirjalili [26] in 2016, which is conducted by imitating whale
preying behavior

I did not know about the WOA algorithm. It was interesting new information for me.
The authors' description is comprehensive and well-written. I appreciate the
detailed formulas.

4) Lines 276-277: In the experiments, 0-230 pieces of data are used as the training
set and 230-330 pieces of data are used as the test set.

What is "a piece of data"?  Is that one record or a set of records? What are the
numbers "0-230" and "230-330" refer to? They do not appear to be mentioned in
Table 2 on p. 7.

5) Lines 304-309: What’s more, we also do another experiment to compare with ARIMA, Linear, Lasso
and SVM. In the experiments, we use our model and the other models to predict the fu-
ture for seven consecutive days which are from August 27 to September 2, 2022. It is
worth mentioning that the training values of the other models are from December 20,
2021 to the day before the forecast. Again, we also use MAE, RMSE, and MAPE to eval-
uate the model effect.

These lines do not belong in the Discussion section. This is a description of a method.
The description of the experiment should be moved into the Materials and Methods. The
results (Fig. 5) should be stated in the Results section, and then discussed in the
Discussion section.

In general, the Introduction should describe and motivate the problem and mention key
related work without critiquing or discussing it. The Materials and Methods section
should simply state the materials (hardware, software, data) and methods (algorithms, models,
experimental design) used in the study without any discussion. The Results section should
simply state the results as obtained facts. It is the Discussion section where the results
should be discussed at length, in and of themselves, and vis-a-vis related work.
For example, statements like:

"However, it can be seen from the overall prediction curve that WOA-BILSTM mod-
el always has the best prediction effect. In addition, with the increase of prediction time,
our model still has a good accuracy."

on lines 293-295 should be moved to the Discussion section.

6) The References section appears to be well formatted according to the MDPI guidelines.
The DOIs are given when appropriate.

7) The Conclusions section is in compliance with the content of the article and does not
overstate the obtained results.

The quality of the English language is acceptable. Some minor modifications are required.

Reviewer 2 Report

Prediction of Covid19 using WOA-BILSTM model

This paper a bulstm combined with WOA to determine the best hyperparaemter to predict covid.

The WOA (Whale Optimisation algorithm is not new, is know since 2016. Its referred but contextualised well. 

You compared the model agaist BILSTM, but this is the base of the proposal with the use of WOA to determine the best set of hyperparameters. There are others ways, such as gaussian processes, such as  Bayesian Hyperparameter Optimization using Gaussian Processes to determine this. How WOA is compared against ?

Trancriving the WOA algorithm is not a good practice, pears summarise it in the key aspects and highlight with the algorithm flow is enough and reference it .

Only 230 parts of data to train, this are instants of time? How this models converged with such little data? What was the batch size??? The loopback window?? 

Who other models were turned? Also using WOA? Grid Search??

In this sentence: They came to the conclusion that models ranking from good performance to the lowest in entire scenarios are Bi-LSTM, LSTM, GRU, SVR and ARIMA. Given Farah's research results and the shortcomings of these deep learning models, we use the BILSTM as our basic model and improve the model with the following changes. There are nowadays newer models that also can achieve better results…

Missing lot of references regarding other works that address same problem using RNN (GRU; LSTM, etc)

The last equation and structure are widely know, perhaps focus only in the cell and main components and the version of bilstm according t0 the diagram presented

While the work seem to be coherence, my concerns mainly include:

1- What is the main novelty?

2- What are the variables to the dataset, what I contains?

3- Is this a multivariate step forecast? Or univariate?

4- How other models were tuned?

5- analysis os sesoonality and ACF and PACF?? 

6- who WOA compares to gaussian process to find hyperparameters

The English is ok, 

English with some errors, but easily fix with a speller.

Reviewer 3 Report

Intetrsting work. I have few remarks:

1. Please provide more details about software and hardware used for the calculations. What were the executions times for these software runs?

2. Are these models capable of predictions of rising epidemic curves only or the decay of epidemic curve is also available?

3. What were the final optimal parameters of the models after WOA optimization? A one example of such optimization would be interesting to show to the Readers

Round 2

Reviewer 2 Report

1- What is the main novelty?

we use the BILSTM as our basic model and improve the model with the following changes. Our approach involves enriching the input data characteristics, including cumulative new, cumulative cured, cumulative deaths, existing infected, daily new, daily cured, and daily deaths. We then use optimization algorithm (WOA) to optimize the model parameters and increase the optimization parameters compared to other optimization models. (We write this on lines 161 -166)

OK, 

2- What are the variables to the dataset, what I contains?

The data set contains 230 training data sets and 100 test data sets, which are explained in lines 321 to 327. We also draw the Figure 4 to explain it.

3- Is this a multivariate step forecast? Or univariate?

It is a multivariate step forecast. I explain this on line 322.

R: its not clear the inout variables, please clarify their significance and value

4- How other models were tuned?

For these models, we set the dropout to 1, the time-step to 7, the optimizer to Adam. And the input data of them is the cumulative number of new cases in the previous period. As for other parameters, we use grid search for optimization.(line 332-334)

Response: Dropout of 1? bsseline models? Any set?

5- analysis os sesoonality and ACF and PACF??

We draw Figure 6 and analyzed it on lines 366 to 371. 6- who WOA compares to gaussian process to find hyperparameters In order to compare the WOA model, we use Bayesian and grid search algorithms to optimize and perform comparative experiments.

Minor edist

Reviewer 3 Report

Dear Authors you still did not provide any description of the software used here

Was it a commercial, open source or in-house written package? Which environment / language was used here?

Round 3

Reviewer 2 Report

Some of my questions were addressed, however the work is not compared with fiision transformer, or multistep transformer for time series.  An quick experiment with same data will highlight the performance of the current apporoach and its main limitations

its ok
